

# Relative importance of tropopause structure and diabatic heating for baroclinic instability

Kristine Flacké Haualand[1] and Thomas Spengler[1]

[1]Geophysical Institute, University of Bergen, and Bjerknes Centre for Climate Research, Bergen, Norway

**Correspondence:** Kristine Flacké Haualand (Kristine.Haualand@uib.no)

**Abstract.** Misrepresentations of wind shear and stratification around the tropopause in numerical weather prediction models can lead to errors in potential vorticity gradients with repercussions for Rossby wave propagation and baroclinic instability. Using a diabatic extension of the linear quasi-geostrophic Eady model featuring a tropopause, we investigate the influence of such discrepancies on baroclinic instability by varying tropopause sharpness and altitude as well as wind shear and stratification in the lower stratosphere, which can be associated with model or data assimilation errors or a downward extension of a weakened polar vortex. We find that baroclinic development is less sensitive to tropopause sharpness than to modifications in wind shear and stratification in the lower stratosphere, where the latter are associated with a net change in the vertical integral of the horizontal potential vorticity gradient across the tropopause. To further quantify the relevance of these sensitivities, we compare these findings to the impact of including mid-tropospheric latent heating. For representative modifications of wind shear, stratification, and latent heating intensity, the sensitivity of baroclinic instability to tropopause structure is significantly less than that to latent heating of different intensities. These findings indicate that tropopause sharpness is less important for baroclinic development than previously anticipated and that latent heating and the structure in the lower stratosphere play a more crucial role, with latent heating being the dominant factor.

## 1 Introduction

The tropopause is characterised by sharp vertical transitions in vertical wind shear and stratification, resulting in large horizontal and vertical gradients of potential vorticity (PV) (e.g., Birner et al., 2006; Schäfler et al., 2020). These PV gradients act as wave guides for Rossby waves and are crucial for their propagation (see review by Wirth et al., 2018, and references therein). Hence, the common notion that tropopause sharpness must be important for midlatitude weather and its predictability (e.g., Schäfler et al., 2018). In addition to the potentially important impact from the structure of the tropopause, baroclinic development is also greatly influenced by diabatic heating associated with cloud condensation (e.g., Manabe, 1956; Craig and Cho, 1988; Snyder and Lindzen, 1991). As diabatic heating strongly influences the horizontal scale and intensification of cyclones (e.g., Emanuel et al., 1987; Balasubramanian and Yau, 1996; Moore and Montgomery, 2004), its misrepresentation is a common source for errors in midlatitude weather and cyclone forecasting (Beare et al., 2003; Gray et al., 2014; Martínez-Alvarado et al., 2016). While the effect of diabatic heating on baroclinic development is relatively well known, few studies have investigated the





impact of tropopause sharpness on baroclinic development. Here, we quantify and contrast these two contributions to baroclinic instability using an idealised framework.

The initialisation of the tropopause in weather and climate prediction models is based on a sparse observational network of satellites and radiosondes, resulting in large estimates of analysis errors and analysis error variance in the tropopause regions (Hamill et al., 2003; Hakim, 2005). Given this challenge in constraining the atmospheric state near the tropopause, it is difficult

to evaluate how potential errors influence the initial state in weather forecast models and thereby the overall predictability of midlatitude weather. In addition to errors related to observations, the representation of tropopause sharpness is further modified by data assimilation techniques (Birner et al., 2006; Pilch Kedzierski et al., 2016). For example, investigating the representation of the tropopause inversion layer, Birner et al. (2006) concluded that data assimilation smoothed the analysis of the sharp vertical temperature gradient just above the tropopause. In contrast, Pilch Kedzierski et al. (2016) found that data

assimilation improved the representation of these sharp gradients, although the gradients were still too smooth and their location displaced from the actual tropopause compared to satellite and radiosonde observations. Given the unclear contributions from data assimilation and observational errors, it remains uncertain how well tropopause sharpness is represented in model analysis and especially how important such a representation is for baroclinic development.

Even if these sharp structures were well represented at the initial state, forecast errors at the tropopause have been shown to

quickly develop in a few days (e.g., Dirren et al., 2003; Hakim, 2005; Gray et al., 2014; Saffin et al., 2017). Using medium-range forecasts from three operational weather forecast centres, Gray et al. (2014) showed that PV gradients at the tropopause were smoothed with forecast lead time due to horizontal resolution and numerical dissipation. One can expect such a smoothing to dominate even more in global climate prediction models due to the coarser resolution. It is, however, unclear how much these forecast errors near the tropopause contribute to forecast errors for midlatitude cyclones.

Another challenge influencing the forecast skill related to structures near the tropopause is the chosen altitude of the top of the atmospheric model, because it affects how artefacts from the upper boundary imprint themselves at the tropopause. Lifting the model lid has been shown to significantly improve the medium-range forecast of the stratosphere (Charron, 2012) as well as climate predictions on intraseasonal to interannual time scales (Marshall and Scaife, 2010; Hardiman et al., 2012; Charlton-Perez et al., 2013; Osprey et al., 2013; Butler et al., 2016; Kawatani et al., 2019). However, no studies have investigated the

direct impact of the model lid on tropopause sharpness. With the discrepancies related to a low model lid potentially affecting the representation of the tropopause, it is valuable to understand how sensitive baroclinic development is to such modifications of the tropopause.

While the modelling challenges related to the model lid, model resolution, data assimilation techniques, and observations typically lead to a smoothing of the sharp PV gradients around the tropopause, they may also contribute to misrepresentations

of wind (Schäfler et al., 2020) and temperature (Pilch Kedzierski et al., 2016) in the stratosphere that result in further deviations in the stratospheric PV gradients. Even if such deviations were small, a change in the difference in wind shear and stratification across a finite tropopause alters the vertical integral of the horizontal PV gradient. For example, increasing the wind in the lower stratosphere, which alters the vertical integral of the horizontal PV gradient by weakening the amplitude of the negative wind shear above the tropopause, influences the nonlinear decay in baroclinic lifecycles (Rupp and Birner, 2021). The authors





also indicated that the linear growth phase of the development might respond more to changes in the stratospheric wind if the horizontal PV gradients were further modified. As no previous studies have directly investigated how modifications in the vertical integral of the horizontal PV gradient influences baroclinic development, the importance of preserving the vertical integral of PV gradients remains unclear.

While tropopause sharpness is mainly related to vertical changes *across* the tropopause, misrepresentations of either strati-
fication or vertical wind shear may also lead to implicit modifications of the altitude of the tropopause itself. Such fluctuations of the tropopause are associated with enhanced analysis and forecast errors (Hakim, 2005) and are often induced by baroclinic waves through vertical and meridional heat transport (Egger, 1995). While some studies argue that baroclinic instability is sensitive to the level of the tropopause (Blumen, 1979; Harnik and Lindzen, 1998), Müller (1991) found that the vertical distance between the waves at the tropopause and at the surface is not very important for baroclinic development. Thus, the net effect
on baroclinic instability by altering stratification and wind shear in ways that affect tropopause altitude remains unclear.

To evaluate the relative importance of the various aspects of tropopause structure and diabatic heating for baroclinic instability, we use a moist extension of the linear quasi-geostrophic (QG) Eady (1949) model where we vary wind shear and stratification across the tropopause using different heating intensities. While previous idealised studies focused on the impact of abrupt environmental changes across the tropopause (e.g., Blumen, 1979; Müller, 1991; Wittman et al., 2007) and how sharp
and smooth transitions across the tropopause affected neutral modes and the longwave cutoff (de Vries and Opsteegh, 2007) as well as wave frequency, energetics, and singular modes (Plougonven and Vanneste, 2010), we systematically investigate the sensitivity of the most unstable baroclinic mode to both changes across the tropopause region as well as different degrees of smoothing. We also include the effect of latent heating and contrast its impact on baroclinic growth to the structure of the tropopause.

## 2   Model and methods

### 2.1   Model setup and solution procedure

Focusing on the incipient stage of baroclinic development, we use a numerical extension of the linear 2D QG model by Eady (1949), formulated similarly to the model of Haualand and Spengler (2019) and Haualand and Spengler (2020), which is based on an analytic version of Mak (1994). We use pressure as the vertical coordinate and assume wavelike solutions in the x
direction for the QG streamfunction $\psi$ and vertical motion $\omega$:

$$[\psi, \omega] = \mathrm{Re}\{ \left[ \hat{\psi}(p), \hat{\omega}(p) \right] \exp(i(kx - \sigma t)) \} ,\tag{1}$$

where the hat denotes Fourier transformed variables, $k$ is the zonal wavenumber, and $\sigma$ is the wave frequency. The non-dimensionalised $\omega$ and potential vorticity (PV) equations can then be expressed as

$$\frac{d^2\hat{\omega}}{dp^2} - Sk^2\hat{\omega} = i2\lambda k^3\hat{\psi} + k^2\hat{Q}\tag{2}$$





and

$$(\overline{u}k - \sigma)\left[\frac{d}{dp}\left(\frac{1}{S}\frac{d\hat{\psi}}{dp}\right) - k^2\hat{\psi}\right] + k\frac{d}{dp}\left(\frac{\lambda}{S}\right)\hat{\psi} = i\frac{d}{dp}\left(\frac{\hat{Q}}{S}\right), \tag{3}$$

where $S$ is the basic-state static stability as defined in Haualand and Spengler (2019), $\lambda$ is the basic-state vertical wind shear, and $\overline{u}$ is the basic-state zonal wind. As introduced by Mak (1994) and implemented by Haualand and Spengler (2019), the diabatic heating rate *divided by pressure* is $Q = -\frac{\varepsilon}{2}h(p)\omega_{lhb}$, where $\varepsilon$ is the heating intensity parameter, $h(p)$ is the vertical

heating profile defined as 1 between the bottom ($p_{lhb}$) and the top of the heating layer ($p_{lht}$) and zero elsewhere, and $\omega_{lhb}$ is the vertical velocity at the bottom of the heating layer.

Unlike Mak (1994) and Haualand and Spengler (2019), we include an idealised tropopause with a default setup of uniform $\lambda$ and $S$ in the troposphere and in the stratosphere, separated by a discontinuity at the tropopause. The discontinuity introduces an interface condition for the vertical integral of the PV equation across the tropopause:

$$\left[\frac{1}{S}\frac{\partial\psi}{\partial p}\right]_{p_*^+}^{p_*^-} \propto \left[\frac{\lambda}{S}\right]_{p_*^+}^{p_*^-}, \tag{4}$$

where $p_*$ is the pressure at the sharp tropopause interface and $p_*^+$ and $p_*^-$ denote locations just below and just above the tropopause, respectively. Following Haualand and Spengler (2019), we refer to $\partial\psi/\partial p$, which is proportional to the negative density perturbation, as temperature. In line with Bretherton (1966), the jump in $\lambda/S$ is proportional to the vertical integral of $\partial\overline{q}/\partial y$ across the sharp tropopause. Thus, the changes in $\lambda$ and $S$ across the tropopause introduce a meridional PV gradient at

the tropopause, which is positive for the parameter space we explore.

The set of equations is completed with the boundary conditions $\hat{\omega} = 0$ at $p_t$ and $p_b$, the thermodynamic equation

$$(\overline{u}k - \sigma)\frac{d\hat{\psi}}{dp} + i\hat{Q} + \lambda k\hat{\psi} = 0 \quad \text{at} \quad p = p_b, \tag{5}$$

as well as $\partial\psi/\partial p = 0$ at $p_t$, where $p_t$ and $p_b$ are the pressure at the top and bottom of the domain, respectively. The upper boundary condition is in line with Müller (1991) and Rivest et al. (1992) and prescribes vanishing temperature anomalies. As

temperature anomalies at the model boundaries can be interpreted as PV anomalies (e.g., Bretherton, 1966; de Vries et al., 2010), this boundary condition is associated with zero PV anomalies at the model top, ensuring that the instability is mainly restricted to the troposphere, where PV anomalies at the tropopause mutually interact with PV anomalies at the surface. Additional tropospheric PV anomalies appear at the top and bottom of the heating layer in the presence of latent heating $Q$.

The default setup is the same as in Haualand and Spengler (2019) with the following exceptions (summarised in Table

1). The tropopause is at $p = p_* = 0.25$, corresponding to 250 hPa, and the model top is, in accordance with Mak (1998), at $p = p_t = 0$. Furthermore, the wind shear $\lambda$ reverses sign across the tropopause, from $\lambda_{tr} = 3.5$ in the troposphere to $\lambda_{st} = -3.5$ in the stratosphere, with the zonal wind profile being defined as

$$\overline{u} = \begin{cases} \lambda_{tr}(p_b - p) & \text{for} \quad p \geq p_* \\ u_* + \lambda_{st}(p_* - p) & \text{for} \quad p < p_* \end{cases}, \tag{6}$$





**Table 1.** Setup of sharp and smooth CTL experiments

|  | $\lambda_{st}\ [\lambda_{tr}]$ | $S_{st}\ [S_{tr}]$ | $p_*$ | $\hat{\alpha}$ | $\delta$ |
|---|---|---|---|---|---|
| nondimensional | -3.5 [3.5] | 4 [1] | 0.25 | 1 | 0.15 |
| dimensional | -0.035 [0.035] | 0.04 [0.01] | 250 | 1 | 150 |
| units | $\mathrm{ms^{-1}hPa^{-1}}$ | $\mathrm{m^2\,s^{-2}\,hPa^{-2}}$ | hPa | . . . | hPa |

$\delta$ only applicable for smooth experiments.

which we argue is a good representation of the zonal wind profile in the midlatitudes when compared to observations (e.g.,
Birner et al., 2006; Houchi et al., 2010; Schäfler et al., 2020). In the stratosphere, the stratification $S_{st} = 4$ remains the same as
that of the full model domain in Mak (1994) and Haualand and Spengler (2019), but is reduced to $S_{tr} = 1$ in the troposphere,
which is a more representative value for the midlatitude troposphere (e.g., Birner, 2006; Grise et al., 2010; Gettelman and
Wang, 2015) and is consistent with previous studies (Rivest et al., 1992; de Vries and Opsteegh, 2007; Wittman et al., 2007).
The choice of a weaker tropospheric stratification results in stronger vertical motion and hence a larger scaling of latent heating
as well as increased growth rates. To compensate for this, we consistently reduce the heating intensity parameter of $\varepsilon = 12.5$
from Haualand and Spengler (2019) to $\varepsilon = 2$, such that the growth rates and the scaling of latent heating remain of the same
order of magnitude as in Haualand and Spengler (2019).

Equations (2), (3), and (5) form an eigenvalue problem that is solved numerically for the eigenvalue $\sigma$ and the eigenvectors
$\hat{\psi}(p)$ and $\hat{\omega}(p)$ for a given wavenumber $k$. Due to the normalization constraint mentioned in Haualand and Spengler (2019),
the eigenvectors $\hat{\psi}(p)$ and $\hat{\omega}(p)$ are scaled arbitrarily and cannot be compared quantitatively across experiments. We use a
numerical resolution of 201 vertical levels with increments of 5 hPa and calculate solutions for 200 different wavenumbers.
See Haualand and Spengler (2019) for further details.

## 2.2 Smoothing procedure

To investigate the sensitivity of baroclinic instability to smoothing the tropopause, we substitute the step function of $\lambda/S$
around the tropopause with a sine function that gradually increases from $(\lambda/S)_{st}$ in the upper stratosphere to $(\lambda/S)_{tr}$ in the
lower troposphere in a vertical range symmetric around the sharp tropopause interface, i.e., $p_* - \delta/2 \leq p \leq p_* + \delta/2$:

$$\frac{\lambda}{S}(p) = \begin{cases} (\lambda/S)_{st}\,\hat{\alpha} & \text{for} & 0 \leq p < p_* - \delta/2, \\ \dfrac{1+\alpha}{2}(\lambda/S)_{tr} + \dfrac{1-\alpha}{2}(\lambda/S)_{tr}\,\sin[\tau(p)] & \text{for} & p_* - \delta/2 \leq p \leq p_* + \delta/2, \\ (\lambda/S)_{tr} & \text{for} & p_* + \delta/2 < p \leq 1, \end{cases} \tag{7}$$

where $\tau(p)$ increases linearly from $-\pi/2$ at $p = p_* - \delta/2$ to $\pi/2$ at $p = p_* + \delta/2$ such that $\sin[\tau(p)] \in [-1, 1]$ for $p \in [p_* -
\delta/2, p_* + \delta/2]$, and $\alpha = \hat{\alpha}\dfrac{(\lambda/S)_{st}}{(\lambda/S)_{tr}}$ is the scaling parameter, with $\hat{\alpha}$ being an offset parameter that shifts $(\lambda/S)_{st}$ such that the
vertical integral of $\partial\overline{q}/\partial y$ around the tropopause region is modified when $\hat{\alpha} \neq 1$ compared to when $\hat{\alpha} = 1$. We conduct sharp



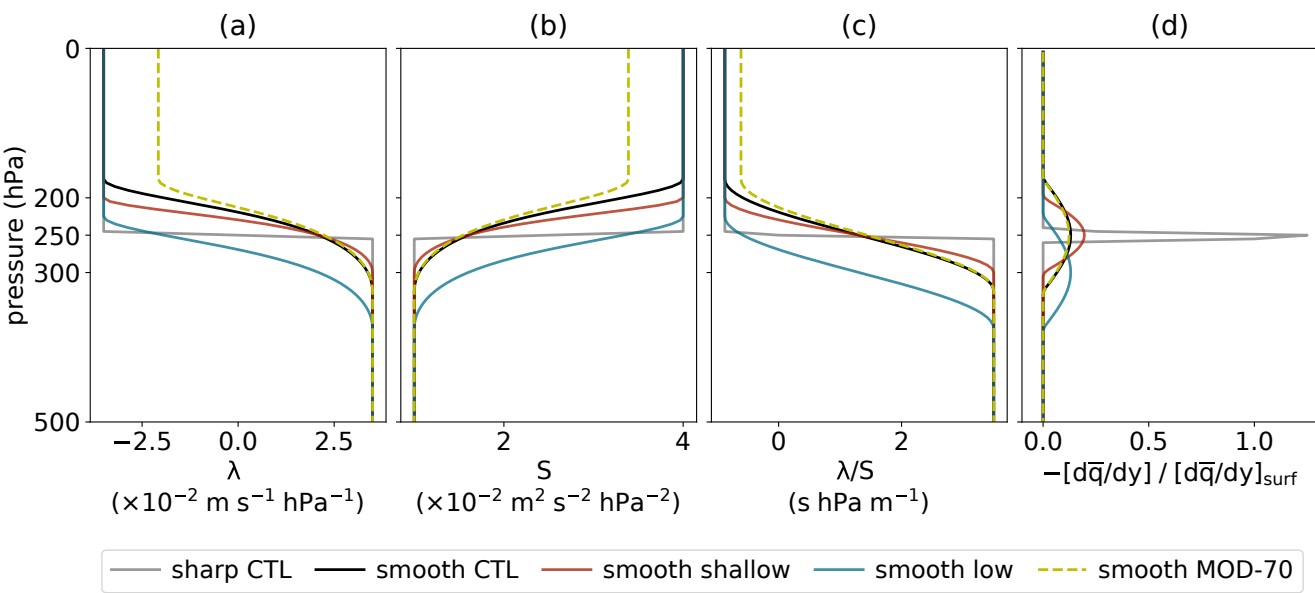

**Figure 1.** Vertical profiles of $\lambda$, $S$, $\lambda/S$, and $\partial\overline{q}/\partial y$ near the tropopause for some key experiments.

and smooth experiments with $p_* = 250$ hPa, $\hat{\alpha} = 1$, and $\delta = 0$ hPa and $\delta = 150$ hPa for the "sharp CTL" and "smooth CTL", respectively (grey and black profiles in Fig. 1). These settings are summarised in Table 1 together with the default setup of $\lambda$ and $S$. We further vary $\delta$ between 50 hPa and 200 hPa (compare black and red profiles in Fig. 1), $p_*$ between 200 hPa and 300 hPa (compare black and blue profiles), and $\hat{\alpha}$ between 1 and 0.7 (compare black and dashed yellow profiles). Note that due

145 to the finite resolution of the model grid, there is always some smoothing even for the sharp profiles, which results in a finite value of $\partial\overline{q}/\partial y$ in Fig. 1d.

The choices for $\delta$, $p_*$, and $\hat{\alpha}$ are based on vertical profiles in the midlatitudes from observational studies (Birner et al., 2002; Birner, 2006; Grise et al., 2010; Gettelman and Wang, 2015; Schäfler et al., 2020), where the motivation for varying the offset parameter down to $\hat{\alpha} = 0.7$ is based on the finding that some models only capture about 70% of the observed magnitude of

150 the wind shear (Schäfler et al., 2020, see their Fig. 9 c,d). Experiments with $\hat{\alpha} \neq 1$, resulting in a modified vertical integral of the horizontal PV gradient, are labeled "MOD", with the offset parameter $\hat{\alpha}$ shown in percentage after "MOD", such that "MOD-70" corresponds to $\hat{\alpha} = 0.7$ and means that $(\lambda/S)_{st}$ is reduced to 70% of its original value. In some cases we also refer to experiments with $\hat{\alpha} = 1$ and hence an unaltered vertical integral of the horizontal PV gradient as "NO-MOD" experiments to avoid confusion with the MOD experiments.

155 After smoothing $\lambda/S$, we define the smoothed profiles of $\lambda$ and $S$ (see Fig. 1a,b) by letting

$$\lambda(p) = \lambda_{st}\hat{\alpha} + \Delta\lambda \cdot \gamma(p) \quad \text{and} \quad S(p) = S_{st}\hat{\alpha} + \Delta S \cdot \gamma(p),$$





where $\Delta\lambda = \lambda_{tr} - \lambda_{st}\,\hat{\alpha}$ and $\Delta S = S_{tr} - S_{st}\,\hat{\alpha}$ are the respective differences in $\lambda$ across the tropopause region and

$$\gamma(p) = \begin{cases} 0 & \text{for} & 0 \le p < p_* - \delta/2, \\[2ex] -\hat{\alpha}\dfrac{\lambda_{st} - S_{st}\dfrac{\lambda}{S}(p)}{\Delta\lambda - \Delta S\dfrac{\lambda}{S}(p)} & \text{for} & p_* - \delta/2 \le p \le p_* + \delta/2, \\[2ex] 1 & \text{for} & p_* + \delta/2 < p \le 1, \end{cases}$$

is a factor based on the smoothed profile of $\lambda/S$ ensuring that the smoothing of $\lambda$ and $S$ is distributed equally from $p = p_* - \delta/2$ to $p = p_* + \delta/2$ relative to the total increments $\Delta\lambda$ and $\Delta S$. After defining the smoothed profile $\lambda(p)$, we set $\overline{u}(p) = \int_{p_b}^{p} \lambda(p)dp$, where we assumed $\overline{u}(p_b) = 0$.

Note that if the step function of $\lambda$ shifts sign at the tropopause, while $S$ is positive everywhere, the zero value of the smoothed profile of $\lambda/S$ will be located at a higher vertical level than the discontinuity of the original sharp profile at $p_*$. Thus, the maximum vertical gradient of $\lambda$ and $S$ is, unlike that of $\lambda/S$, typically shifted above the tropopause (compare e.g., black lines in Fig. 1a-c).

## 2.3 Energy equations

The relation between baroclinic growth and changes in wind shear and stratification across the tropopause is investigated from the energetics perspective following Lorenz (1955). The tendency of domain averaged eddy available potential energy EAPE is

$$\frac{\partial}{\partial t}(\text{EAPE}) = C_a - C_e + G_e, \tag{8}$$

where $C_a = -\dfrac{\lambda}{S}\overline{\psi_x \psi_p}$ is the conversion from basic-state available potential energy (APE) to EAPE, $C_e = \overline{\omega\psi_p}$ is the conversion from EAPE to eddy kinetic energy, and $G_e = -\dfrac{1}{S}\overline{Q\psi_p}$ is the diabatic generation of EAPE. The bar denotes zonal and vertical averages.

## 2.4 Validity of QG assumptions

Although several other studies have implemented discontinuous vertical profiles of $\lambda$ and/or $S$ around an idealised tropopause in QG models (e.g., Robinson, 1989; Rivest et al., 1992; Juckes, 1994; Plougonven and Vanneste, 2010), Asselin et al. (2016) argued that the quasi-geostrophic approximation is less appropriate near sharp gradients and narrow zones like the tropopause. Hence, to justify our modelling framework, we tested the validity of the QG approximation by comparing the magnitude of the QG terms in the thermodynamic equation with the magnitude of the nonlinear vertical advection term neglected in the QG framework. As such a quantitative comparison between linear and nonlinear terms requires a scaling of variables (see section 2.1), we chose a maximum surface wind of $5\,\text{ms}^{-1}$ across all experiments, which is in line with our focus on the incipient stage of baroclinic development.





For the sharp CTL experiment, where profiles are discontinuous across the tropopause, the nonlinear vertical advection term is less than 0.25 of the dominant QG term in the thermodynamic equation at all grid points in the baroclinic wave apart from the tropopause interface (not shown). Given the discontinuity at the tropopause due to the jump in wind shear and stratification, the temperature is actually undefined at this level and it is therefore inconsistent to evaluate the thermodynamic equation at this interface.

For the smooth CTL experiment, where profiles are smoothed across the tropopause, the vertical advection term is also less than 0.25 of the dominant QG term at most grid points, though near the tropopause this ratio becomes up to 7.5 (4.7) [3.3] when the vertical extent of the tropopause is 100 (150) [200] hPa. Thus, there are grid points where the non-linear vertical advection term becomes dominant. However, that we obtained qualitatively similar solutions for all smoothing ranges, including the sharp experiment, indicates the suitability of QG framework to explore the sensitivity to the sharpness of the tropopause.

## 3 Impact of wind shear and stratification across the tropopause on baroclinic growth

### 3.1 Control setup with sharp jet and stratification jump

Introducing the effect of variations in $\lambda$ and $S$ across the tropopause, we first compare the sharp CTL experiment, where both $\lambda$ and $S$ are discontinuous across the tropopause (see Sect. 2.1), with setups where either only $\lambda$ is discontinuous across the tropopause (sharp CTL-$\lambda$) or only $S$ is discontinuous across the tropopause (sharp CTL-S). For the sharp CTL experiment, the growth rate [wavelength] of the most unstable mode (black line in Fig. 2) is stronger [longer] than if only $\lambda$ is discontinuous (grey) and weaker [shorter] than if only $S$ is discontinuous (blue). For all of these experiments, there is a longwave cutoff that is related to a non-matching phase speed of the waves at the tropopause and the surface, which is in line with the arguments by Blumen (1979), de Vries and Opsteegh (2007), and Wittman et al. (2007). The qualitative differences in growth rate and wavelength of the most unstable mode as well as the shortwave and longwave cutoffs between these three experiments are the same as those found by Müller (1991) (see his Fig. 2). We present a more detailed discussion of these findings in subsection 3.2, where we explore the parameter space of $\lambda$ and $S$ more extensively.

Below the tropopause, the structure of $\psi$ (shading in Fig. 3a) and temperature $T$ (black contours) for the most unstable mode is similar to the structure of the most unstable Eady mode, with $\psi$ tilting westward and $T$ tilting eastward with height. Together with the westward tilt in both $\omega$ (Fig. 3a) and meridional wind $v = ik\psi$ (not shown, but phase shifted a quarter of a wavelength upstream from $\psi$), this structure is baroclinically unstable and is consistent with warm air ascending poleward and cold air descending equatorward.

In contrast to the Eady model, where the tropopause is represented by a rigid lid, the inclusion of a tropopause with discontinuous profiles of $\lambda$ and $S$ introduces nonzero $\omega$ at the tropopause interface. Just below the tropopause, this nonzero $\omega$ adiabatically cools (warms) the air upstream of the positive (negative) temperature anomaly (compare grey contours and shading in Fig. 3a), thereby weakening the temperature wave as well as accelerating its downstream propagation. This effect is opposed by the meridional temperature advection, which warms (cools) the air upstream of the positive (negative) temperature anomaly just below the tropopause. Thus, with a negative meridional temperature gradient associated with the positive



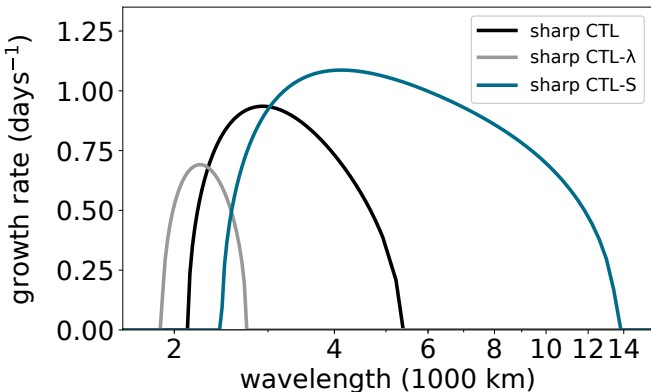

**Figure 2.** Growth rate vs. wavelength for the sharp CTL (black), CTL-$\lambda$ (grey), and CTL-S (blue) experiments. See text for further details.

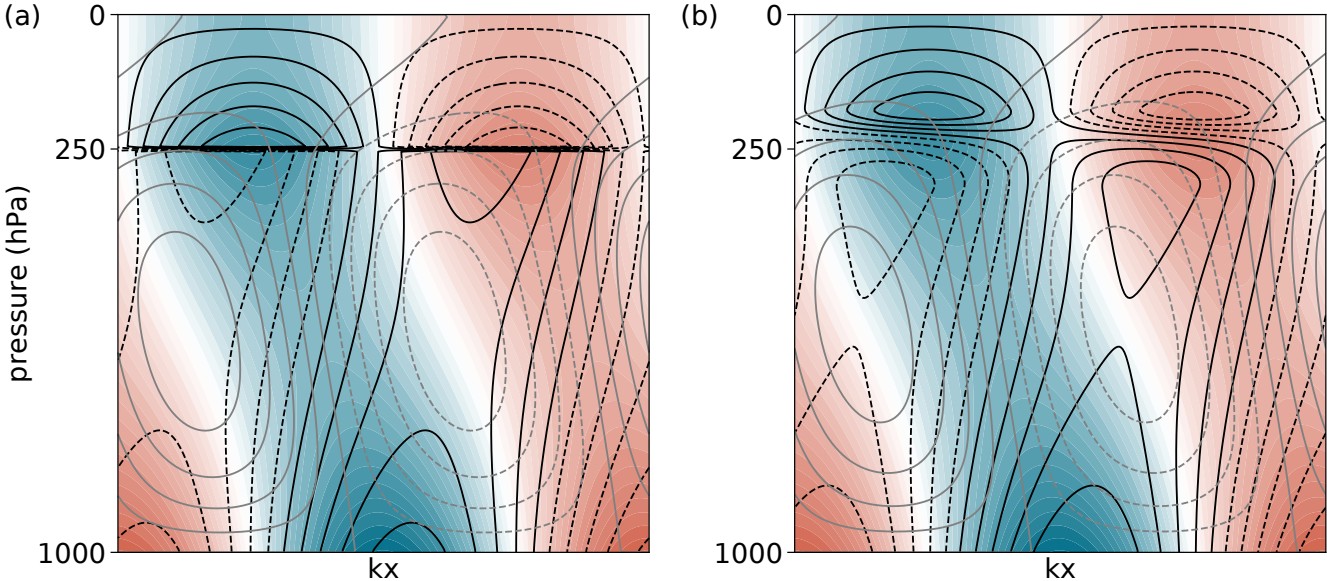

**Figure 3.** Structure of $\psi$ (shading), $-\partial\psi/\partial p$ (black contours), and $\omega$ (grey contours) for (a) the sharp CTL experiment and (b) the smooth CTL experiment. Values are not comparable to physical values due to normalisation constraint mentioned in Sect. 2.1.

wind shear $\lambda$ via the thermal wind relation, meridional temperature advection amplifies the temperature wave and retards its downstream propagation at this level. The net effect is propagation against the zonal wind such that the propagation speed of the temperature wave just below the tropopause matches the propagation speed of the wave at the surface. Only when these propagation speeds are identical, the waves can phase lock and travel together with a common propagation speed that equals
the average phase speed of the two waves (de Vries and Opsteegh, 2007).





The phase of the temperature wave reverses across the tropopause and does not tilt with height in the entire stratosphere (shading in Fig. 3). Such a barotropic structure is in line with the lack of mutual intensification of PV anomalies in this layer. There is a monotonic decay of the temperature anomaly toward the top of the model domain related to the upper boundary condition $\partial\psi/\partial p = 0$. Together with the barotropic structure, this decay yields $T \propto -\partial\psi/\partial p$ being exactly in phase with $-\psi$

and therefore also exactly 90 degrees out of phase with $v = ik\psi$. Nevertheless, due to the reversal of the wind shear across the tropopause, the meridional temperature advection is still retarding the downstream wave propagation above the tropopause such that the stratospheric part of the wave propagates together with the tropospheric part.

However, due to the 90 degrees phase shift between $v$ and $T$, meridional advection can no longer amplify the stratospheric part of the temperature wave. Instead, the amplification of the wave in the stratosphere is entirely due to $\omega$, where $\omega$ is almost

in phase with temperature. Hence, the role of $\omega$ on the amplification of the wave reverses across the tropopause.

The weakening and acceleration of the temperature wave just below the tropopause associated with nonzero $\omega$ is in line with a weaker growth rate, higher phase speed, and hence longer wavelength compared to the most unstable Eady mode (compare contour at the black dot with the black contour in Fig. 4). Such effects on baroclinic development were also found in similar experiments by Müller (1991) and partly by de Vries and Opsteegh (2007).

**3.2 Sensitivity to variations in stratospheric wind shear and/or stratification**

Varying $\lambda_{st}$ and $S_{st}$ while holding $\lambda_{tr}$ and $S_{tr}$ fixed changes $\partial\overline{q}/\partial y$ through its relation to the jump in $\lambda/S$ across the tropopause (see Eq. (4) and related arguments), which has implications for baroclinic growth through the arguments of mutual intensification by interacting PV anomalies (Hoskins et al., 1985). For the parameter space explored in this study, decreasing $\lambda_{st}$ relative to $\lambda_{tr}$ always increases $\partial\overline{q}/\partial y$, whereas increasing $S_{st}$ relative to $S_{tr}$ increases $\partial\overline{q}/\partial y$ only when $\lambda_{st}$ is positive

and decreases $\partial\overline{q}/\partial y$ when $\lambda_{st}$ is negative (Fig. 4d).

The increase in $\partial\overline{q}/\partial y$ for varying $\lambda_{st}$ and $S_{st}$ yields the observed decrease in phase speed and wavelength (compare pattern of black contours in Fig. 4b-d). As argued by Wittman et al. (2007), the relation between $\partial\overline{q}/\partial y$, phase speed, and wavelength is in line with the proportionality of the phase speed of Rossby waves to $-1/k \cdot \partial\overline{q}/\partial y$. Thus, a larger positive $\partial\overline{q}/\partial y$ reduces the phase speed, which can be partly compensated by increasing the wavenumber $k$. A similar qualitative relation between

increasing wavelengths for decreasing $\partial\overline{q}/\partial y$ related to varying $\lambda_{st}$ and $S_{st}$ was found by Müller (1991) (see his Fig. 2b). Müller (1991) also found that decreasing $\lambda_{st}$ reduces the phase speed for a ratio of static stability of 1.5 across the tropopause (see his Fig. 3a-c), which is confirmed by our results (Fig. 4c). Furthermore, our results also show that this relation between $\lambda$ and phase speed holds for all investigated configurations of $S_{st}$.

The sensitivity on the growth rate is less straightforward, with growth rates being largest in the upper right corner of the

$\lambda$-$S$ parameter space, where the wind shear is uniform and the stratification in the stratosphere is larger than in the troposphere (Fig. 4a). Growth rates decrease from this maximum toward weaker $\lambda_{st}$ and $S_{st}$. A similar sensitivity on the growth rate to changes in $\lambda$ and $S$ was found by Müller (1991), where the growth rate of the most unstable mode also peaked when $\lambda_{st}$ and $S_{st}$ were large and decreased toward weaker $\lambda_{st}$ and $S_{st}$ (see his Fig. 2a). While the decrease in growth rates toward the upper left corner of the $\lambda$-$S$ parameter space in Fig. 4a can be explained by the absence of a tropopause due to a uniform $\lambda$ and $S$


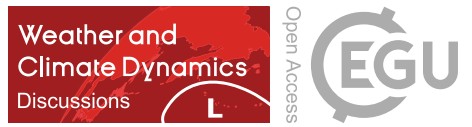

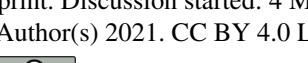

**Figure 4.** Growth rate, wavelength, and phase speed of the most unstable mode together with absolute value of $\partial\overline{q}/\partial y$ at the tropopause relative to its value at the surface for various $\lambda$ and $S$ in the stratosphere (subscript st) and troposphere (subscript tr). Black (yellow) contours show absolute values of experiments with discontinuous (smooth) profiles, and shading shows relative difference between the discontinuous and smooth experiments in percentage. The values for the most unstable Eady mode with a rigid lid at the tropopause using $\lambda_{tr}$ and $S_{tr}$ are marked by a bold contour. Small black dots indicate regions where no solution is calculated due to the absence of unstable solutions. Big black, grey, and blue dots mark the configurations of $\lambda$ and $S$ used for the sharp CTL, CTL-$\lambda$, and CTL-S experiments in Fig. 2, respectively.

**Figure 5.** Same as Fig. 4, but for $-\partial\psi/\partial p$ and $\psi$ at the tropopause relative to surface and the phase shift of $T$ and $v$ at the surface and just below the tropopause.

resulting in no upper level wave and hence no instability, the relation of the growth rate to the choices in the $\lambda$-$S$ parameter space is more complex.

To further understand the changes in growth rate, we consider the conversion of basic-state APE to EAPE ($C_a$), which is constant with height in the troposphere where PV anomalies mutually intensify (not shown). As this energy conversion term





is the main source for EAPE when dry baroclinic waves intensify, it should reflect the observed changes in growth rate. We

therefore explore this term by considering the location and amplitude of $v \sim \psi_x$ and $T \sim \psi_p$.

Just below the tropopause, $v$ and $T$ are more in phase when $\lambda_{st}$ is positive (Fig. 5d), which is beneficial for the energy conversion. At the surface, $v$ and $T$ are generally less in phase than just below the tropopause and changes in phase between $v$ and $T$ are small for different $\lambda_{st}$ and $S_{st}$ (Fig. 5c). Given that $C_a$ is constant throughout the troposphere, the different phase relation between $v$ and $T$ at the surface and just below the tropopause are consistent with larger amplitudes of $v = ik\psi$ and $T$

at the surface relative to the amplitudes just below the tropopause (Fig. 5a,b). This dominance of $v$ and $T$ at the surface relative to just below the tropopause is strongest when the phase between $v$ and $T$ just below the tropopause and the magnitude of $\partial \overline{q}/\partial y$ are small (compare pattern of Figs. 4d, 5a, and 5d). For positive $\lambda_{st}$, we thus argue that the beneficial phase relation between $v$ and $T$ and the larger amplitudes of $v$ and $T$ at the surface favour a larger conversion of basic-state APE to EAPE ($C_a$) compared to when $\lambda_{st}$ is negative. With a large source of EAPE, baroclinic growth is expected to intensify.

To justify the argument relating increased growth rates to an increased source of EAPE through $C_a$, we need to understand what sets the phase relation between $v$ and $T$. Due to the difference condition for temperature across the tropopause in Eq. (4), where the difference in $1/S \cdot \partial \psi/\partial p$ is proportional to the jump in $\lambda/S$ and hence the vertical integral of $\partial \overline{q}/\partial y$ across the tropopause, the temperature anomaly typically reverses across the tropopause (see example in Fig. 3a). When the jump in $\lambda/S$ is large, the temperature difference is also large, such that the temperature anomaly just below the tropopause becomes

zonally more aligned with the opposite temperature anomaly just above the tropopause, reducing the freedom for a phase shift to a more beneficial phase relation with the meridional wind. In contrast, when the jump in $\lambda/S$ is small, the difference in temperature across the tropopause is less constrained such that the temperature anomaly just below the tropopause can more easily be shifted upshear to be more in phase with the meridional wind.

In line with these arguments, the jump in temperature across the tropopause is monotonically increasing with decreasing

$\lambda_{st}/\lambda_{tr}$ when $S_{st}$ and $S_{tr}$ are constant (not shown). In contrast, as mentioned in the beginning of this subsection, an increase in $S_{st}$ relative to $S_{tr}$ increases the jump in $\lambda/S$ only when $\lambda_{st}$ is positive and is therefore not always associated with an increase in the difference of $T$ across the tropopause. Furthermore, as $S$ appears on both sides of the difference condition in Eq. (4), an increase of $S_{st}$ relative to $S_{tr}$ can compensate for a significant part of the changes in the jump of $1/S \cdot \partial \psi/\partial p$, which would leave the temperature more or less unaltered.

It is also worth noting that increasing $S_{st}$ yields a more dominant omega term in the thermodynamic equation that amplifies the temperature anomaly just above the tropopause (as discussed in section 3.1). For a given difference in temperature across the tropopause, the latter effect allows the temperature wave below the tropopause to move more freely away from its antiphase relation with the wave above the tropopause, thereby improving its correlation with $v$. The above arguments related to the complex role of $S$ on temperature near the tropopause demonstrate that the phase relation between $v$ and $T$ just below the

tropopause is more sensitive to changes in $\lambda$ than $S$ (as shown in Fig. 5d).

The arguments related to the beneficial phase relation between $v$ and $T$ for large $\lambda_{st}$ together with the absence of instability for uniform $\lambda$ and $S$, i.e., no tropopause, yield the observed pattern in growth rates (Fig. 4a), with a maximum where $\lambda$ is uniform and the jump in $S$ is large. Hence, baroclinic growth is not largest when the tropopause is at its most abrupt



configuration (lower right corner around the black dot in Fig. 4), but rather when the linear increase in zonal wind is extended

to above the tropopause (upper right corner around the blue dot in Fig. 4).

## 4   Impact of smoothing the tropopause on baroclinic growth

### 4.1   Sensitivity to variations in stratospheric wind shear and/or stratification

Smoothing the vertical profiles of $\lambda$ and $S$ in a vertical extent of 150 hPa around the tropopause yields a similar structure of the most unstable mode as for the experiments with discontinuous profiles (compare Figs. 3a and b). Moreover, the sensitivity to $\lambda$

and $S$ for growth rate, wavelength, phase speed, and $\partial \overline{q}/\partial y$, as well as the amplitude and phase of $v$ and $T$ remain qualitatively the same after smoothing (compare black and yellow contours in Figs. 4 and 5), with growth rates still peaking when $\lambda_{st}$ and $S_{st}$ are large.

Even though smoothing weakens the maximum of $\partial \overline{q}/\partial y$ by 90% (shading in Fig. 4d), the growth rate, wavelength, and phase speed change by less than $\pm 4\%$ (shading in Fig. 4a-c). In line with a weaker $\partial \overline{q}/\partial y$ and the dispersion relation for

Rossby waves (as discussed in Sect. 3.2), smoothing increases the wavelength and the phase speed for most of the investigated configurations of $\lambda_{st}$ and $S_{st}$ (Fig. 4b,c) and decreases the growth rates by up to 2.9% when $\lambda_{st}$ is negative and $S_{st}$ is weak (Fig. 4a).

However, when $\lambda_{st}$ and $S_{st}$ are large, the growth rate *increases* by up to 0.9% (Fig. 4a). We argue that this enhancement is related to an improved phase relation between $v$ and $T$ compared to the experiments with discontinuous profiles (shading

in Fig. 5d), where a smooth tropopause with a wider vertical distribution of $\partial \overline{q}/\partial y$ yields more flexibility in relative location between the temperature anomalies just below and above the tropopause. Such an improved phase relation is associated with enhanced conversion of basic-state APE to EAPE and may overcompensate for the detrimental impact from the weakening of $\partial \overline{q}/\partial y$. In fact, for the most realistic setup where both $\lambda$ and $S$ change across the tropopause (around the black dot in Fig. 4a), the sensitivity on the growth rate from smoothing is almost negligible, indicating that the positive impact related to the

improved phase relation between $v$ and $T$ is balanced by the detrimental impact from the weakening of $\partial \overline{q}/\partial y$. This suggests that baroclinic growth is typically not very sensitive to an accurate representation of $\lambda$ and $S$ around the tropopause.

The perhaps largest qualitative difference from the impact of smoothing on the overall instability analysis is an additional mode at long wavelengths when $\lambda_{st}$ is negative and $S_{st}$ is large (Fig. 6). The streamfunction structure of this mode features its strongest westward tilt with height within the smoothed tropopause region and decays rapidly above (not shown). This mode

exists only due to the additional levels of opposing and nonzero $\partial \overline{q}/\partial y$ in the smoothed tropopause region. We will not focus on these modes at long wavelengths, as we argue that their weak growth rate and long wavelength as well as their westward tilt bound solely to the tropopause region make them less relevant for an assessment for typical midlatitude cyclones.



**Figure 6.** (a) Growth rate vs. wavelength for $\lambda_{tr} = 3.5$, $\lambda_{st} = -3.5$, $S_{tr} = 1$, $S_{st} = 4$ and various sensitivity experiments, with the default smooth experiment being associated with a tropopause region of 150 hPa depth centered at an altitude of 250 hPa. See text for further details. (b) Zoom-in of (a).

## 4.2 Sensitivity to vertical extent and altitude of tropopause

Comparing the sensitivity of baroclinic growth to the vertical extent of smoothing, tropopause height, and changes in the
vertical integral of $\partial \overline{q}/\partial y$ (see details in Sect. 2.2), the greatest sensitivity is related to the changes in the vertical integral of $\partial \overline{q}/\partial y$, where the growth rates of the sharp and smooth MOD-70 experiments are similar and increase by 2.7% to 3.5%




compared to their NO-MOD counterpart experiments (compare sharp and faint colours in Fig. 6). The increase in growth rate
from the NO-MOD experiments to the MOD-70 experiments is associated with a decrease in $\partial\overline{q}/\partial y$ at the tropopause toward a
more optimal value that better matches with the $\partial\overline{q}/\partial y$ at the surface (not shown), such that the waves at the tropopause and at
the surface can more easily phase lock and travel together with the same phase speed (Blumen, 1979; de Vries and Opsteegh,
2007; Wittman et al., 2007). For the NO-MOD experiments, the sensitivity to tropopause height (solid and dashed blue in Fig.
6) and vertical extent of smoothing (solid and dashed red) changes the growth rate by only -0.24% to 0.31% compared to the
sharp control experiment (black).

The sensitivity to vertical extent of smoothing and tropopause height is qualitatively the same for both the NO-MOD and the
MOD-70 experiments. Lowering (raising) the tropopause weakens (enhances) the growth rate (solid and dashed blue in Fig.
6). This can be related to an increased (decreased) vertical average of $S$ from the surface to the tropopause (Fig. 7)

$$\overline{S}_{tr} = \frac{1}{p_* - p_b} \int\limits_{p_b}^{p_*} S_{tr} \, dp,$$

where the stratification is related to the growth rate through the inverse proportionality between the static stability and the
maximum Eady growth rate (Lindzen and Farrell, 1980; Hoskins et al., 1985).

In contrast to the sensitivity to tropopause height, increasing the vertical extent of smoothing does not necessarily have a
monotonic impact on the growth rate. Deepening the tropopause region from a narrow (solid red in Fig. 6) to an intermediate
(dash-dotted black) vertical extent of smoothing increases the growth rate. However, deepening the tropopause further from
an intermediate to a wide (dashed red) extent of smoothing barely changes the growth rate. Moreover, when increasing the
smoothing further, i.e., beyond the displayed sensitivity range, the growth rate starts to decrease (not shown). For the MOD-70
experiments, the turnover point, i.e., where increased extent of smoothing starts to weaken the growth rate, exists at a larger
extent of smoothing that is beyond our sensitivity range considered for the NO-MOD experiments (not shown).

The maximum in growth rate for some intermediate degree of smoothing is associated with an intermediate $\partial\overline{q}/\partial y$ and
an intermediate phase speed of the wave at the tropopause (recall that the phase speed for Rossby waves is proportional to
$-\partial\overline{q}/\partial y$). Such an intermediate phase speed appears to be the most optimal phase speed yielding the best match in phase speed
for the surface wave, such that the waves at these two levels phase lock and intensify each other as efficiently as possible.

Changes in growth rate relative to the sharp CTL experiment are summarised in Fig. 8, including experiments with simulta-
neous modifications of the vertical extent and altitude of the tropopause for different modifications of the vertical integral of
the PV gradient. This figure highlights that the main relative change in growth rate is related to the modification of the vertical
integral of the PV gradient rather than modifications of vertical extent and altitude of the tropopause.

**4.2.1 Changes in growth rate and corresponding forecast error**

The changes in growth rate may seem small, but as variables grow nearly exponentially at the incipient stage of develop-
ment, errors grow quickly with time. Relative to a reference experiment (subscript *ref*), the forecast error of the relative wave



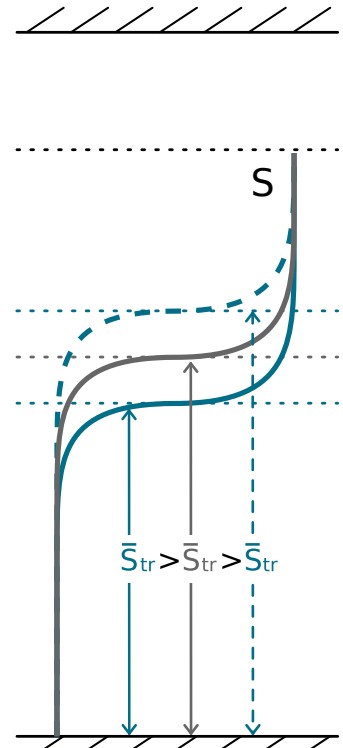

**Figure 7.** Schematic illustrating how the altitude of the tropopause modifies the vertical average of the tropospheric stratification $\overline{S}$.

amplitude $A'/A_{\text{ref}}$ at the time $t = t_1$ is

$$\left.\frac{A'}{A_{\text{ref}}}\right|_{t=t_1} = \left(1 + \left.\frac{A'}{A_{\text{ref}}}\right|_{t=0}\right) \exp[(\sigma - \sigma_{\text{ref}})t] - 1. \tag{9}$$

Assuming perfect initial conditions, i.e., $A'/A_{\text{ref}} = 0$ at $t = 0$, the forecast error for the NO-MOD smooth experiments relative to the sharp control experiment is less than +/- 1% [2%] during a short-range forecast of 2 days [medium-range forecast of 5 days], while the corresponding error for the MOD-70 experiments is up to 6% [17%] (dashed lines in Fig. 9). In comparison, assuming a relative initial error of 5%, the relative forecast error is down to 4% [3%] after 2 [5] days for the NO-MOD smooth experiments, and up to 12% [22%] for the MOD-70 experiments. The decrease in the relative error for some of the NO-MOD

smooth experiments is a result of an underestimate of the growth rate relative to the sharp control experiment, which reduces the initial positive relative error. If the growth rates are compared to the growth rate of a weakly smoothed experiment instead of the sharp reference experiment, the error is more or less unaltered. We therefore let the growth rate of the sharp experiment be the reference for the error growth calculations.

      Keeping in mind that these results are based on a highly idealised model, the findings indicate that it is not so important

if models fail to accurately represent $\lambda$ and $S$ around the tropopause. Instead, it is much more important that $\lambda$ and $S$ are well represented in the lower stratosphere, such that the vertical integral of $\partial \overline{q}/\partial y$ around the tropopause region is preserved.





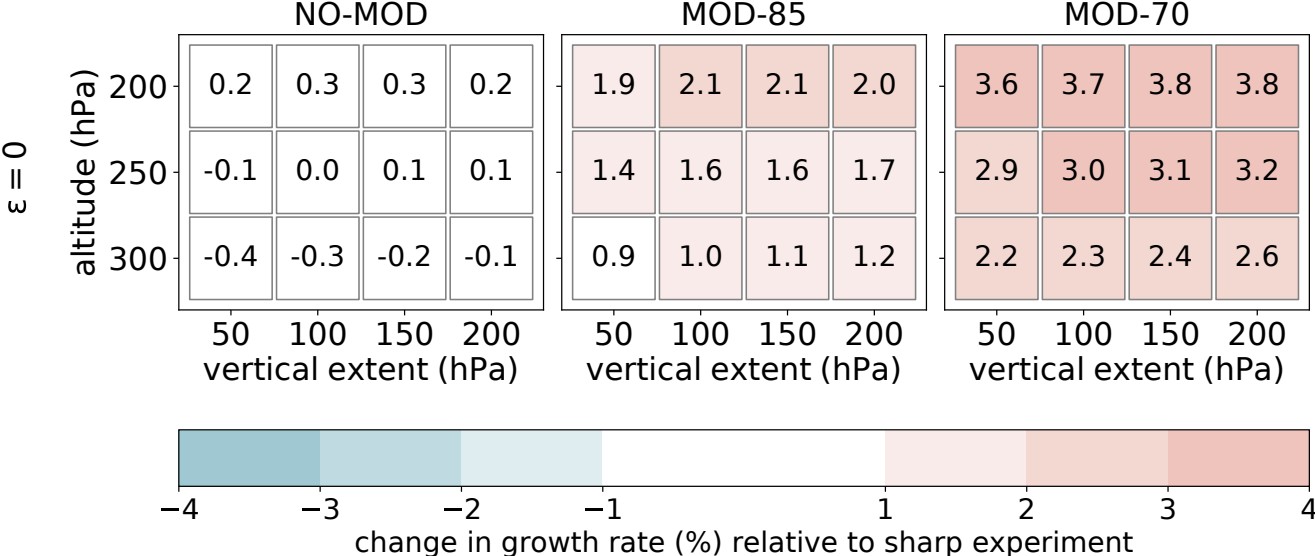

**Figure 8.** Change in growth rate (shading and numbers) for various smooth experiments relative to CTL with discontinuous profiles from Fig. 6 for no latent heating ($\varepsilon = 0$).

The importance of representing the lower stratospheric winds is further supported by Rupp and Birner (2021), who found that baroclinic lifecycle experiments are sensitive to changes in the wind structure in the lower stratosphere. Such changes in wind structure are often related to a downward extension of a weak polar vortex after sudden stratospheric warming events (Baldwin
and Dunkerton, 2001), which have been shown to significantly alter midlatitude weather in the troposphere (see review by Kidston et al., 2015, and references therein).

### 4.3  Sensitivity to latent heating intensity

Including latent heating in the mid-troposphere does not significantly change the qualitative findings of the sensitivity experiments from section 4.2 (compare Fig. 10 with Fig. 6). Nevertheless, the most unstable mode at shorter wavelengths is
associated with dominant diabatic PV anomalies at the heating boundaries (Fig. 11b), which align with the westward tilt of $\psi$ (Fig. 11a). Growth rates peak at shorter wavelengths, which is consistent with the presence of diabatic PV anomalies and hence a shallower effective depth of interacting PV anomalies (Hoskins et al., 1985).

For some of the experiments, the weak and positive growth rates at long wavelengths are split into two modes (Fig. 10). The longest of the two is similar to their adiabatic counterpart mentioned in the end of Sect. 4.1, while the shortest of the two is
associated with the increased dominance of the diabatic PV anomalies at the top of the heating layer. Due to the irrelevance for midlatitude cyclones mentioned in section 4.1, these modes are beyond the scope of this study.

In line with the dominance of diabatic PV anomalies in the lower and middle troposphere, latent heating also weakens the relative sensitivity to the modifications of the vertical integral of $\partial \overline{q}/\partial y$ across the tropopause (compare Fig. 10 with Fig. 6),



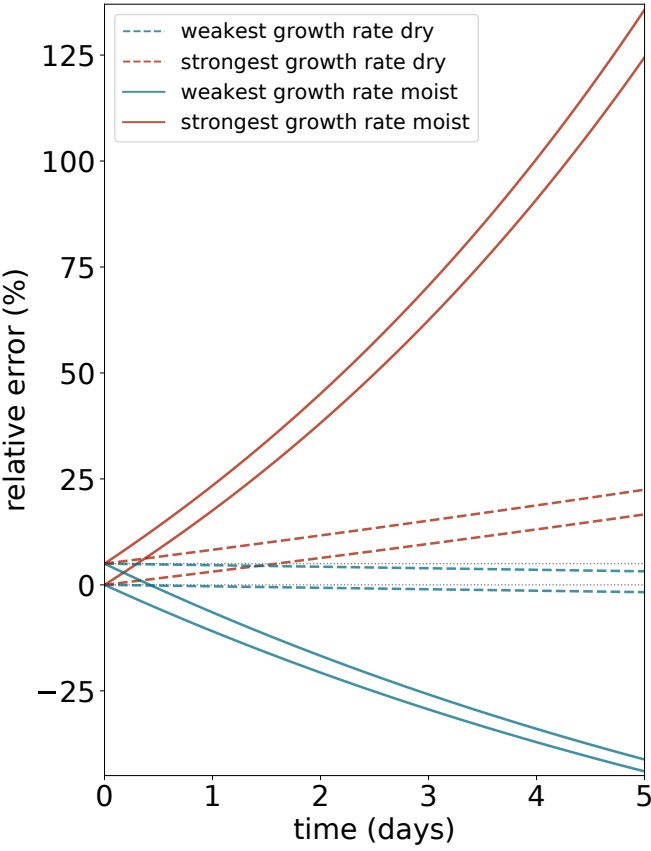

**Figure 9.** Evolution of error for the weakest (blue) and strongest (red) maximum growth rates from Fig. 6 (dashed, dry) and Fig. 10 (solid, moist) starting with initial relative errors of 0% and 5% (grey dotted horizontal lines).

with growth rates for the MOD-70 experiments increasing by only 1.0-1.1% relative to the NO-MOD counterpart experiment
instead of 2.7-3.5% as for the adiabatic experiments. Keeping the idealised context of this study in mind, this finding indicates
that the presence of latent heating makes models relatively less vulnerable to an inaccurate representation of $\lambda$ and $S$ around
the tropopause.

Decreasing (increasing) the heating parameter from $\varepsilon = 2$ to $\varepsilon = 1.5$ ($\varepsilon = 2.5$), which corresponds to a 25% decrease (in-
crease) in latent heating and associated precipitation, yields a much larger variation in the maximum growth rate compared to
the tropopause sensitivity experiments for a fixed heating parameter (Fig. 12). The change in growth rate relative to the sharp
experiment for $\varepsilon = 2$ is between -10.2% (for $\varepsilon = 1.5$) and +14.2% (for $\varepsilon = 2.5$), and the corresponding error after 2 [5] days is
between -21% [-44%] (for $\varepsilon = 1.5$) and +38% [+124%] (for $\varepsilon = 2.5$) if there are no initial errors, and a few percent larger if
the relative initial error is 5% instead (solid lines in Fig. 9). In comparison, the corresponding numbers for the relative change
in growth rate when changing the latent heating intensity $\varepsilon$ by only 5% [10%] instead of 25% are between -2.4% [-4.5%] and
+2.4% [+4.9%] instead of -10.2% and +14.2%.







**Figure 10.** Same as Fig. 6 but including latent heating with $\varepsilon = 2$.

All aforementioned changes associated to the intensity of the diabatic heating are larger than the relative changes in growth rate for the various tropopause smoothing experiments for a fixed $\varepsilon = 2$ (middle row in Fig. 12), which range between -0.2% and +1.7%. Moreover, these findings remain similar when using smooth vertical profiles of latent heating as in Haualand and Spengler (2019) (see their Fig. 11a), with the relative change in growth rate being between -5.0% and +3.0% when changing the latent heating intensity $\varepsilon$ by 5% (not shown). Again, these numbers are all larger than the change in growth rate relative to the experiment with the discontinuous profiles for a fixed $\varepsilon = 2$, which are between -2.1 and +1.9% when using a smooth



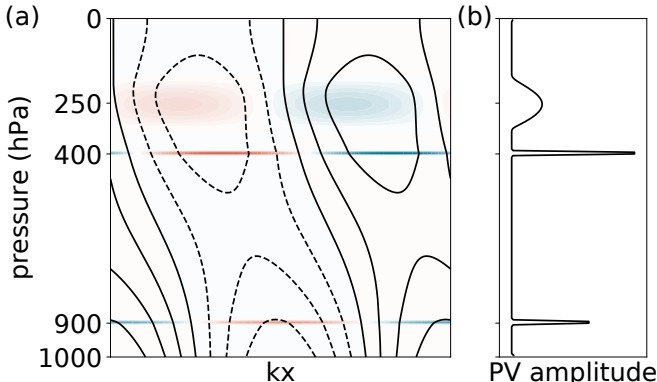

**Figure 11.** (a) Structure of interior PV (shading) and $\psi$ (contours) for the most unstable mode of the default smooth experiment including latent heating and (b) amplitude of interior PV anomalies. Values are not comparable to physical values due to normalisation constraint mentioned in Sect. 2.1.

heating profile. With such a high sensitivity of the forecast error to heating intensity, our results indicate that it is much more important to adequately represent diabatic processes than the sharpness of the tropopause.

## 5 Conclusions

Including sharp and smooth transitions of vertical wind shear and stratification across a finite tropopause in a linear QG model extended from the Eady (1949) model, we investigated the relative importance of changes across the tropopause region at different degrees of smoothing on baroclinic development and compared its sensitivity to that of diabatic heating. We found that impacts related to tropopause structure are secondary to diabatic heating related to mid-tropospheric latent heating.

In contrast to the Eady mode, where the tropopause is represented by a rigid lid, the inclusion of an idealised tropopause with
abrupt changes in wind shear and/or stratification introduces nonzero vertical motion at the tropopause. The vertical motion leads to adiabatic cooling/warming at the tropopause, which opposes the effect of meridional temperature advection. The adiabatic cooling/warming weakens the amplitude of the wave at the tropopause but accelerates its downstream propagation, resulting in weaker growth rates and higher phase speed than the most unstable Eady mode.

In agreement with the dispersion relation for Rossby waves, increasing (decreasing) $\partial \bar{q}/\partial y$ at the tropopause by varying
the stratospheric wind shear and/or stratification is associated with relatively weak (strong) phase speed and short (long) wavelength. In contrast to wavelength and phase speed, the impact from wind shear and stratification on the growth rate is less straight forward, with growth rates being strongest when wind shear is uniform and the increase in stratification is large across the tropopause. The strong growth rates are related to a beneficial phase relation between meridional wind and temperature near the tropopause, which is associated with enhanced conversion of basic-state available potential energy to eddy available
potential energy. Thus, baroclinic growth is not strongest when the tropopause is sharpest.





**Figure 12.** Same as Fig. 8, but including latent heating for three different heating intensity parameters ($\varepsilon = 1.5, 2.0, 2.5$). Note that the colorbar is extended from the one in Fig. 12 but contains the same colours at lower values.

Smoothing the tropopause is associated with a positive effect on baroclinic growth related to a further enhancement of energy conversion through an improved phase relation between meridional wind and temperature, as well as a negative effect related to a weaker maximum gradient of $\partial \overline{q}/\partial y$ in the tropopause region. The positive effect from smoothing dominates when there are no or small changes in wind shear and large changes in stratification across the tropopause, resulting in increased growth

rates compared to when the tropopause is sharp. In contrast, the negative effect dominates when there are large changes in





wind shear and no or small changes in stratification, yielding weaker growth rates than for a sharp tropopause. For the most realistic configuration, with large changes in both wind shear and stratification across the tropopause, these opposing effects balance each other, resulting in negligible changes in growth rate from smoothing, suggesting that baroclinic growth is not very sensitive to tropopause sharpness.

The effect of smoothing for a realistic configuration of wind shear and stratification remains weak when increasing the vertical extent of smoothing and altering the tropopause altitude, with an error growth for exponentially growing quantities of less than 2% in a medium-range forecast of 5 days. In contrast, modifying the wind shear and stratification above the tropopause, resulting in modifications in vertical integral of the PV gradient relative to a sharp control experiment, has a much more pronounced effect on baroclinic growth than the effects related to smoothing and varying tropopause altitude.

The associated exponentially growing forecast error of any wave amplitude assuming perfect initial conditions is 17% in a medium-range forecast of 5 days when the stratospheric wind shear divided by stratification is reduced to 70% of its original value, which is a reduction actually occurring in operational numerical weather prediction models (Schäfler et al., 2020). The relatively large sensitivity to the lower stratospheric winds on baroclinic development is in line with Rupp and Birner (2021), who also argued that baroclinic growth may be sensitive to modifications in the horizontal PV gradients.

Although the relative impact on baroclinic growth depends on how much the profiles of wind shear and stratification are altered for the different sensitivity experiments, our estimates indicate that it is much more important to maintain the vertical integral of the PV gradient than to accurately represent the abrupt vertical contrasts across the tropopause. Such modifications above the tropopause may represent modelling challenges related to observational errors, vertical resolution, a low model lid, or limitations related to data assimilation techniques, but they can also represent changes in the lower stratospheric winds

resulting from downward extensions of a weak polar vortex after a sudden stratopheric warming event.

As expected from the strong impact of diabatic heating on baroclinic development, including mid-tropospheric latent heating of moderate intensity increases the growth rate. However, including latent heating does not alter the qualitative findings regarding the impact of tropopause structure on baroclinic development. Nevertheless, modifying the heating intensity by 5-25% has a significantly larger impact on the growth rate than the effects of smoothing tropopause structure, varying tropopause altitude,

and maintaining the vertical integral of the PV gradient. This highlights the main finding of this study that baroclinic growth is more sensitive to diabatic heating than tropopause structure.

While this study is the first to quantify the relative effect of tropopause sharpness and latent heating on baroclinic development, it is important to keep in mind the highly idealised character of this study. More realistic simulations with numerical weather prediction models should be performed to test our findings and to further clarify the relative importance of the repre-

sentation of the tropopause and diabatic forcing on midlatitude cyclones.

*Code availability.* The current version of the model code is available on https://github.com/krifla/2dQGnum/tree/v1.0.0. A digital object identifier (DOI) will be provided if this work gets published in WCD.



*Author contributions.* KFH and TS designed the experiments and KFH carried them out. KFH developed the model code, and KFH and TS analysed the model output. KFH prepared the manuscript with support from TS.

*Competing interests.* The authors declare that they have no conflict of interest.

*Acknowledgements.* We thank Michael Reeder for his valuable input on an earlier version of the manuscript and Vicky Meulenberg for her preliminary work on tropopause sharpness during her internship in Bergen which motivated this study. The work has been carried out within the Research Council of Norway project UNPACC (RCN project number 262220).





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
