# Peer review of "Relative importance of tropopause structure and diabatic heating for baroclinic instability"

_Weather and Climate Dynamics, 2021_

## Author Response (AR1)

**Response to Reviewer 1 for wcd-2021-13**

**We thank the reviewer for reading through our manuscript and appreciate the concise and constructive review. Below we respond to each comment and mark the response with bold font.**

This is an interesting paper containing usefuly insights into the relative role of shear and sharpness at the tropopause. It uses an idealised framework for the investigation, which as the authors acknowledge would require further investigation. This should not be seen as a criticism however as the idealised setting provides the ideal setting for testing ideas and forming hypetheses to be tested in a more comprehensive setting as long as limitations are clearly articulated.

**We are very happy to see that the reviewer acknowledges the value of the idealised framework.**

I feel the following points need to be addressed.

Comments:

Ln 10:
Last line of the abstract. This is slightly too strongly worded. Suggest including the word "may": "These findings may indicate that tropopause sharpness is less important for baroclinic development than previously anticipated and that latent heating and the structure in the lower stratosphere may play a more crucial role, with latent heating being the dominant factor."

**This is a fair point that we will address in the revised manuscript.**

Ln 185: Is it true that that temperature cannot be defined, or is it just the definition is arbitrary (e.g. like zero point of heaviside function). Can it not be defined as the limit as a smooth tropopause tends to a discontinuity or a matching condition for equations above and below the discontinuity. This is a minor point.

**We thank the reviewer for raising this point. We agree that temperature can be arbitrarily defined at the tropopause level. In the model in use, the model output includes a streamfunction with a breaking point at the tropopause, such that the vertical derivative of it, i.e., temperature, is a priori discontinuous and undefined at the tropopause. In the postprocess, we have chosen to define tropopause temperature as the average of the temperature just above and just below the tropopause, which is probably the best arbitrary definition. Such a definition should be consistent with, or at least similar to, the temperature in the limit where a smooth tropopause becomes a step function. We will modify the statement in line 185 in the revised manuscript to make this clear.**

Ln 190: "However, that we obtained qualitatively similar solutions for all smoothing ranges, including the sharp experiment, indicates the suitability of QG framework to explore the sensitivity to the sharpness of the tropopause." The rationale here is not clear to me. How does consistency within the QG framework imply consistency in a more comprehensive setting? This needs to be explained more clearly or perhaps an acknowledgement that this is a limitation of the work included.

**We agree with the reviewer that this is a limitation that needs to be acknowledged. Despite the caveat of self-consistency, we believe that the robustness of the results across solutions for a wide**

**range of a smoothing and different dominance of the nonlinear vertical advection term supports the QG framework. We will, however, make sure to clarify these limitations in the revised manuscript.**

Ln 210: How is the non-zero vertical velocity and consequent advection across a discontinuous tropopause justified? Surely this would lead to raising, sinking of the tropopause level. In the Eady model this is avoided by enforcing zero omega at the rigid lid. In the idealised setting discontinuous heating profiles are usually assumed to represent a change in state of the moisture - e.g. the lifting condensation level. What is the rationale for maintainence of the sharp tropopause in the present work? Simply small amplitude perturbations?

**We appreciate the comment about providing a rationale for nonzero vertical motion at the tropopause. Our arguments that the shift of the tropopause level due to vertical motion is a minor effect that can be neglected in this idealised framework are twofold. Firstly, given our focus on the incipient stage of development, a feedback on the tropopause by the perturbations is small. Secondly, vertical motion is significantly reduced at the tropopause compared to the mid-troposphere.**

Edits in the manuscript based on comments from Reviewer 1 (also highlighted in the track-changes file):

Ln 10:

These findings indicate that tropopause sharpness might be less important for baroclinic development than previously anticipated and that latent heating and the structure in the lower stratosphere could play a more crucial role, with latent heating being the dominant factor.

Ln 185:

For the sharp CTL experiment, where profiles are discontinuous across the tropopause, the nonlinear vertical advection term is less than 0.25 of the dominant QG term in the thermodynamic equation at all grid points in the baroclinic wave apart from the tropopause interface (not shown). Given the discontinuity at the tropopause due to the jump in wind shear and stratification, the temperature is a priori undefined at this level. Evaluating the thermodynamic equation with an arbitrary definition of temperature at this interface would therefore be inconsistent.

Ln 190:

Thus, there are grid points where the non-linear vertical advection term becomes dominant. With the uncertain implications of such a dominance for our findings, the validity of the QG framework should be further tested in more comprehensive models accounting for the nonlinear vertical advection term. Nevertheless, that we obtained qualitatively similar solutions for all smoothing ranges, including the sharp experiment, indicates the suitability of QG framework to explore the sensitivity to the sharpness of the tropopause.

Ln 210:

In contrast to the Eady model, where the tropopause is represented by a rigid lid, the inclusion of a tropopause with discontinuous profiles of $\lambda$ and S introduces nonzero $\omega$ at the tropopause interface. Such a nonzero vertical motion would in reality lead to undulations in the tropopause interface that cannot be represented by this linear framework. However, focusing on the incipient stage of development, a feedback on the tropopause by the perturbations is small. Furthermore, vertical velocities are significantly reduced at the tropopause compared to the mid-troposphere yielding a minor effect on the wave structure that can be neglected during the linear phase of the growth of the perturbation.

Just below the tropopause, the nonzero $\omega$ adiabatically cools (warms) the air upstream of the positive (negative) temperature anomaly (compare grey contours and shading in Fig. 3a), thereby weakening the temperature wave as well as accelerating its downstream propagation. This effect is opposed by the meridional temperature advection, which warms (cools) the air upstream of the positive (negative) temperature anomaly just below the tropopause.

**Response to Reviewer 2 for wcd-2021-13**

**We thank the reviewer for reading through our manuscript and appreciate the concise and constructive review. Below we respond to each comment and mark the response with bold font.**

The authors explore different representations of the tropopause transition in basic state variables and their role in baroclinic normal mode growth in an Eady-like setup including a stratosphere. Specifically, vertical shear and static stability are implemented either as step-like vertical profiles or with a smoother transition from tropospheric to stratospheric values. It is found that smoothing this tropopause transition in one or both of these variables has generally small effects on baroclinic growth. Furthermore, these effects are mostly non-trivial. What matters most appears to be the vertically integrated basic state meridional PV gradient, which depends to some degree on the specifics on the modified setups. These results are contrasted with the effects due to tropospheric diabatic heating, which are found to be much more important than the representation of tropopause sharpness.

The results are interesting and help to clarify questions related to misrepresentations of certain processes and structures in numerical models. Use of a highly idealized setup has the advantage of being able to allow quantitative mechanistic insights, but as usual comes at the price that it remains an open question how the results carry over to complex models and/or the real atmosphere. Overall the presentation is clear and I don't see any major objections to publication, although I do have some general as well as specific comments that I hope will help the authors improve their presentation and discussion of results.

**We are happy to see that the reviewer appreciates the results and the idealised nature of the study.**

General comments:

1. Comparison between the effects of tropopause sharpness and latent heating is useful. But I think a cleaner distinction would help: essentially, tropopause sharpness represents a modification of the already existing basic state PV gradient structure whereas latent heating introduces new PV gradient structures. So even just intuitively it seems more likely for the latter to have a more significant effect due to the stronger, more qualitative modification of the basic state. So, contrasting both effects one-to-one may be a bit "unfair".

**We agree with the reviewer that it is not fair to directly compare modifications in the PV structure around the tropopause with the introduction of new PV structures in the mid troposphere from diabatic processes. For this reason, we investigate the sensitivity to latent heating intensity by using a moderate heating intensity (middle row in Fig. 12) as a reference and compare it to cases with weaker or stronger heating intensity (upper and lower rows in Fig. 12, respectively). This way we make sure that the modifications in latent heating intensity are reasonable compared to the modifications in near-tropopause structure. The justification for the realism of the modifications in near-tropopause structures is stated in lines 147ff, whereas a similar justification for the +/- 25% modifications in latent heating intensity is indicated in lines 393-394. The uncertainty of comparable modifications in heating intensity is further elaborated by reducing the modifications from a 25% change in intensity to a 5% change in lines 398-400.**

**Before we compare the modifications in heating intensity with modifications in tropopause structure, we also discuss how the sensitivity of baroclinic growth to tropopause structure changes when we go from no heating to moderate heating. This part is essential for understanding why the role of the tropopause changes slightly when heating is included. Nevertheless, based on the reviewer's comment, it might not be clear enough in the original manuscript that it is not this *introduction* of heating - but rather the *modification* of heating of moderate intensity - that is used as a comparison to the sensitivity to tropopause structure. We will clarify this in Section 4.3 of the manuscript.**

2. Do you think the results on tropopause sharpness would change much for the case of non-zero interior PV gradient? One reason they might is that with the classic Eady setup the tropopause PV gradient always stands out compared to the zero interior gradient regardless of its strength. Once the interior gradient is non-zero this may change the general picture. I admit I'm not sure what specific changes to expect but I'd be curious to hear the authors' thoughts on this question.

**We are happy to discuss this point of non-zero interior PV gradients. In the current model version, interior PV gradients exist when there are vertical changes in wind shear and stratification, which is mainly near the tropopause. Thus, the smoothing of the tropopause already introduces nonzero PV gradients in the upper troposphere and lower stratosphere. In addition, several previous studies (see e.g., Vallis, 2006) have added the beta effect in Eady-like models and found that the resulting interior PV gradients typically weaken the growth rate and make the structure more surface-based, because any vertical level in the interior needs to interact with the surface. With the enhanced role of the surface, we anticipate that tropopause sharpness and its associated PV gradients may become even less important than without these interior PV gradients.**

**Vallis, G. K., 2006. Atmospheric and oceanic fluid dynamics, p. 274-277. Cambridge University Press.**

3. It could be helpful to include some comments about the role (or lack thereof) of strength of tropopause PV gradient on vertically propagating stationary waves (see Lindzen and Roe, 1997 which is a correction to the earlier Lindzen, 1994).

**We agree that this study by Lindzen and Roe addresses aspects that are of potential interest for the broader context of our paper. However, as their study focuses on the effect on stationary waves, where modifications in the PV structure change the PV gradient in the entire troposphere, we argue that it is not directly relevant for our work. To keep the context of our study concise and clear, we therefore decided beforehand that we won't refer to this study. We hope that this decision sounds reasonable.**

Specific comments:

line 91: I think most readers would prefer if you copy the expression for S here (and for PV would be helpful, too)

**This is a fair point that we will implement in the revised version of the manuscript. We will also make sure to point to the definition of QG PV, which is the expression inside the square brackets of Eq. (3).**

line 100, Eq. 4: please discuss how the proportionality is to be applied (i.e., do you need to introduce a proportionality factor?)

**To be clearer about the proportionality between the left and right hand side of the equation, we decided to expand this equation in the revised version of the manuscript.**

Fig. 1, caption: I think it would help to spell out the meaning of the parameters and briefly describe the different experiments (similar comment applies to other Fig. captions)

**We thank the reviewer for pointing this out. The experiments are mentioned in lines 140-146 in the text, but we agree that they should also be briefly mentioned in the caption. We will fix this in the revised version. We will also extend the captions of Fig. 2 and Fig. 8, but decided to not repeat the description of all the sensitivity experiments in the figure captions and instead refer to the definitions in the text.**

line 198: matter of taste, but I think readability would be improved if you formulate an extra sentence for the wavelength results (i.e., avoid the short-form expressions with square brackets; similar comment applies to other places)

**This is indeed a matter of taste of either a dense formulation or a slightly repetitive formulation. We decided to implement the suggestion from the reviewer in the revised manuscript.**

line 260: notation for derivatives is changed here, I suggest using consistent notation

**We are glad the reviewer spotted this. We will make sure the notation is consistent throughout the manuscript.**

lines 457ff: I agree that it's important to end the paper with this caveat. Another caveat is that this paper only considers the growth phase of baroclinic instability and that the results reported here may not carry over to the mature and decay phases, even in idealized settings. I encourage the authors to include a related comment (and perhaps emphasize at a few places throughout the paper that they only consider this one stage of the life cycle).

**We agree that our results are probably not very applicable to later stages of development which are typically highly nonlinear. In the original manuscript, we mentioned the focus on the incipient stage of development three places (lines 82, 181, 356). Nevertheless, we agree that we could be clearer about these limitations and decided to also mention it in the last paragraph of the conclusions.**

Edits in the manuscript based on comments from Reviewer 2 (also highlighted in the track-changes file):

Ln 393ff, edit based on general comment #1:

To compare the sensitivity of baroclinic growth to modifications in heating intensity with the sensitivity to modifications in tropopause structure, we decrease (increase) the heating parameter from $\varepsilon = 2$ to $\varepsilon = 1.5$ ($\varepsilon = 2.5$), which corresponds to a 25% decrease (increase) in latent heating and associated precipitation. Such modifications in heating intensity yield a much larger variation in the maximum growth rate compared to the tropopause sensitivity experiments for a fixed heating parameter (Fig. 12).

Ln 92:

...where QG PV is defined by the expression inside the square brackets, $S=-R/p \, (dT_0/dp - R \, T_0/c_p \, p)$ is the basic-state static stability with R being the gas constant, $c_p$ being the specific heat at constant pressure, and $T_0$ being the background temperature, $\lambda$ is the...

Ln 100:

$[1/S \, d\psi/dp] = -k \, \psi(p\_*) / (uk-\sigma) \, [\lambda/S] \sim [\lambda/S]$

Caption for Figure 1:

Vertical profiles of $\lambda$, S, $\lambda/S$, and $\partial q/\partial y$ near the tropopause for the sharp and smooth control experiments (grey and black, respectively) contrasted with experiments featuring a smooth shallow tropopause with delta=100 hPa (red), a smooth low tropopause with $p\_*$=300 hPa (blue), and a reduction of stratospheric wind shear divided by stratification to 70% of the original value, i.e., hat{alpha}=0.7 (yellow).

Caption for Figure 2:

Growth rate vs. wavelength for the sharp CTL (black), CTL-$\lambda$ (grey), and CTL-S (blue) experiments, where either lambda and S, only lambda, or only S are discontinuous, respectively.

Caption for Figure 8:

Change in growth rate (shading and numbers) for various smooth experiments relative to the CTL experiment with the same discontinuous profiles of lambda and S from Fig. 6 for no latent heating ($\varepsilon = 0$). See text for further details.

Lns 197-199:

For the sharp CTL experiment, the growth rate of the most unstable mode (black line in Fig. 2) is stronger than if only $\lambda$ is discontinuous (grey) and weaker than if only S is discontinuous (blue), while the wavelength of the most unstable mode is longer than if only $\lambda$ is discontinuous and shorter than if only S is discontinuous.

Ln 260 and lns 171-173:

*Notation for derivatives updated for consistency.*

Lns 457ff:

While this study is the first to quantify the relative effect of tropopause sharpness and latent heating on baroclinic development, it is important to keep in mind the highly idealised character of this study, which limits the focus of the study to the incipient stage of development. More realistic simulations with numerical weather prediction models should be performed to test our findings and to further clarify the relative importance of the representation of the tropopause and diabatic forcing on midlatitude cyclones.